# Speckled Tracking of Pleura—A Novel Tool for Lung Ultrasound; Distinguishing COVID-19 from Acute Heart Failure

**DOI:** 10.3390/jcm11164846

**Published:** 2022-08-18

**Authors:** Batsheva Tzadok, Yair Blumberg, Moti Shubert, Majdi Halabi, Eran Tal-Or, Noa Bachner-Hinenzon, Shemy Carasso

**Affiliations:** 1Baruch Padeh Medical Center, Poriya 1528001, Israel; 2Azrieli Faculty of Medicine, Bar-Ilan University, Zefat 1311502, Israel; 3Department of Cardiology, Rivka Ziv Medical Center, Zefat 1008, Israel; 4Sanolla, Nesher 3688519, Israel

**Keywords:** speckled tracking, B protocol, COVID-19 pneumonia, congestive heart failure, lung ultrasound, viral pneumonia, point-of-care ultrasound, COVID-19, emergency medicine COVID-19

## Abstract

For the acutely dyspneic patient, discerning bedside between acute decompensated heart failure (ADHF) and COVID-19 is crucial. A lung ultrasound (LUS) is sensitive for detecting these conditions, but not in distinguishing between them; both have bilateral B-lines. The Blue protocol uses pleural sliding to differentiate decreased pneumonia; however, this is not the case in ADHF. Nonetheless, this pleural sliding has never been quantified. Speckled tracking is a technology utilized in the echocardiography field that quantifies the motion of tissues by examining the movement of ultrasound speckles. We conducted a retrospective study of LUS performed in emergency room patients during the COVID-19 pandemic. Speckled tracking of the pleura by applying software to the B-mode of pleura was compared between COVID-19 patients, ADHF patients, and patients with no respiratory complaints. A significant difference was found between the patient groups on speckled tracking both in respect of displacement and velocity. ADHF had the highest displacement, followed by COVID-19, and then non-respiratory patients: 1.63 ± 1.89, 0.59 ± 0.71, and 0.24 ± 0.45, respectively (*p* < 0.01). A similar trend was seen in velocity with ADHF having the highest velocity 0.34 ± 0.37, followed by COVID-19 0.14 ± 0.71, and non-respiratory patients 0.02 ± 0.09 (*p* <0.01). Speckled tracking of the pleura is a potential tool for discerning between different causes of dyspnea.

## 1. Introduction

Increased extravascular lung water (EVLW) is a fluid accumulation in the interstitial and alveolar spaces. It is a phenomenon that has varying etiologies [1]. The main causes of EVLW in the critically ill are acute decompensated heart failure (ADHF), acute respiratory distress syndrome (ARDS), and infectious pneumonia [1]. In the acutely dyspneic patient, the ability to discern between these etiologies within minutes may be the determining factor between the patient living or dying [2]. Since the onset of the COVID-19 pandemic, this need has become more crucial than ever. The consequences of mistaking a COVID-19 patient for an ADHF patient not only compromise the patient’s treatment, but also the hospital infection control measures [3].

LUS, as a diagnostic modality, entered the intensive care literature in 1993 [4]. It uses artifact alterations originating from the pleural line, called B-lines, to detect lung EVLW [5]. LUS has been found to have a sensitivity above 90% for detecting ADHF [6]. LUS has also been shown to outperform the sensitivity of chest X-rays in detecting pneumonia [7].

Differentiating between the causes of EVLW using LUS is a challenge; this is because ADHF, pneumonia, and ARDS all have B-lines on imaging. Lichtenstein’s Blue protocol [5] and ensuing works claimed they could differentiate between these pathologies based on the presence of or decreased absent pleural lung sliding [8,9]. ARDS and pneumonia with a disrupted pleura have a decreased pleural sliding; however, ADHF with an intact pleura has an increased pleural sliding. It is of note that pleural sliding in these fundamental works is based on the observations of expert sonographers; it is problematic when the use of LUS is extended to doctors at different levels of expertise. This dilemma brings to light the need to develop a tool that may quantify pleural movement.

With the onset of the COVID-19 pandemic, we saw an increased use of LUS for diagnostics [10,11]. LUS proved itself as a tool both for prognostication as well as monitoring disease progression [12,13]. The major LUS findings were B lines and confluent, consolidations, pleural line thickening, and irregularity [14]; however, a reduced movement of pleural was not reported as a finding. Yet, due to the similarities in pathophysiology of ARDS, pneumonia, and COVID-19 [15], we would expect to see in COVID-19 a decrease in lung sliding.

Speckled tracking is a method developed for echocardiography. It uses the movement of ultrasound speckles to provide both quantitative and qualitative information regarding tissue deformation and motion. It has been shown to be an effective tool in determining the strain of the left ventricle on echocardiography [16]. While speckled tracking in echocardiography may be used to appreciate the multiple vectors of cardiac strain, in a lung ultrasound it may be used for the detection or quantification of pleural sliding, and the movement of parietal on visceral pleura. Two recent articles found speckled tracking of the pleura to be more sensitive than a B-mode ultrasound in detection of pneumothorax [17,18]. The quantification of pleural sliding via speckled tracking in other lung pathologies, to our knowledge, has never been studied.

A recent study of LUS in an emergency department patient population presenting with suspected COVID-19 reached a sensitivity of up to 94%, yet a specificity of only 7–13% [19]. This portrays the challenges we have as physicians in diagnosing COVID-19 in a relatively older, sicker patient population, who may have confounding factors on LUS. Our hypothesis was that the pleural displacement, measured by speckled tracking, should be significantly different between ADHF and COVID-19 patients; this leads to the ability to discriminate between the two conditions reliably and consistently. Since the pleura is not contracting, we assumed that its displacement represents respiratory exertion; this measurement might have a clinical value. The aim of our study was therefore to determine if speckled tracking of the pleura is discriminatory between COVID-19 versus ADHF.

## 2. Materials and Methods

### 2.1. Subjects and Study Design

We conducted a retrospective study of ultrasounds collected in the emergency department of the Poriah-Padeh medical center, one of the major government hospitals in Northern Israel. The study was approved by the institutional ethical board (ethical number: 0113–19). The need for informed consent was waived by the institutional board as the study was a retrospective review of images for diagnostic purposes. The ultrasounds were collected between April 2020 and May 2021 during the COVID-19 pandemic in Israel. The ultrasounds were collected as a convenience sample of all patients presenting to the adult emergency department in this period.

### 2.2. Collection of Ultrasound Clips

The ultrasound was performed by a single physician. The anterior chest was observed with a linear probe, and the posterior and axillary regions with a curvilinear probe. The transmission frequency of the probes was 4–12 Hz and the curvilinear was 6–2 Hz. The frame rate of the ultrasound cines was 18–30 F/m. There was no protocol of marking lung zones; the lung was covered in a “lawnmower” fashion from the superior aspect of the thorax to the lung base [20]. Abnormalities were recorded and saved in a clip with the identification number of the patient. Recordings were done with a Mindray Te-7 ultrasound machine (Shenzen Mindray Bio-Medical Electronics Co Ltd, Itamar St. Petach Tikvah, Israel); one patient’s images were obtained by GE Logiq (Eldan Electronic Instruments Ltd, 6 Hashiloach St. Petach Tikva, Israel).

### 2.3. Classifying Patient Groups

Patients’ records were reviewed by two physicians both with over 5 years of experience, and were classified into three groups: acute decompensated heart failure, COVID-19 pneumonia, and no respiratory complaint. The classifications were based on the presenting complaint, course of hospitalization, PCR of COVID-19, pro-BNP, and chest X-ray if performed. The patients were considered COVID-19 if they had a positive PCR test; the patients were considered ADHF if there was an elevated pro-BNP and clinical diagnosis of ADHF on hospital discharge.

### 2.4. Analysis of B Mode

The images in B mode were examined by two physicians: one with over three years’ experience with lung ultrasounds and one with over ten years of experience. Both physicians were blinded to the patient’s condition. They were asked to determine if there was a smooth or unsmooth pleura, and if the pleura moved.

Following this, the pleura was examined using speckled tracking software free trace option (Siemens Velocity vector imaging, VVI, Mountain View, CA, USA) applied to B-mode ultrasound clips. It is of note that a COVID-19 lung and ADHF have areas of unaffected lung next to the affected lung. To ensure that the pathology was represented, the areas chosen for speckled tracking were areas showing B-lines or consolidated B-lines in the ADHF and COVID-19 cohort. The region of interest was marked manually in B mode (Figure 1) and analyzed by the software (Figure 2) please refer to Appendix A.

### 2.5. Statistical Analysis

We described the main characteristics and comorbidities, as well as the main complaints with which the patients came to the ED.

All the results are presented as mean ± SE. Analysis was performed using SPSS software (IBM SPSS Statistics for Windows, Version 25.0. Armonk, NY, USA); one-way analysis of variance was used to determine the difference between the displacement of the pleura during the birthing cycle in ADHF, COVID-19, and non-respiratory patients.

The Kendall rank correlation coefficient was used to determine the agreement between the physicians where <0.3 was considered a weak correlation.

## 3. Results

A total of 208 ultrasounds were analyzed: 60 COVID-19 patients, 66 ADHF patients, and 82 patients with non-respiratory complaints. The demographics are presented in Table 1. The ADHF cohort was older than the COVID-19 and non-respiratory patient group (Table 1).

There was a significant difference in the displacement between all three groups, with ADHF having the most displacement, followed by COVID-19, and a non-respiratory abnormality: 1.63 ± 1.89, 0.59 ± 0.71, 0.24 ± 0.45 mm (*p* < 0.01) (Figure 2 and Figure 3). In addition, the ADHF showed a significant increased velocity with comparison to COVID-19 and non-respiratory abnormalities, 0.34 ± 0.37, followed by COVID-19 0.14 ± 0.71, and non-respiratory patients 0.02 ± 0.09 Cm/s (*p* < 0.01) (Figure 4). Based on our results, a larger displacement and rapidly moving pleura is more indicative of ADHF; whereas a slow-moving pleura with less displacement is indicative of COVID-19. There was no difference within each subgroup if the images were obtained with a linear or curvilinear probe (*p* value > 0.1).

The B-mode analysis showed a significant difference between the smoothness of the pleura, with a weak agreement between the two physicians (Kendall rank correlation coefficient = 0.256)

There was no agreement between the physicians as far as pleural movement, with poor interrelated agreement (Kendall rank correlation coefficient = 0.132).

## 4. Discussion

Dyspnea comprises up to 20% of patients presenting to emergency departments [21]. With the aging of our patient population, we find ourselves less able to depend on the classic signs and symptoms to diagnose our patients [22]. This may be seen in our study with respect to the major complaints of patients (Table 1): pedal edema, commonly presented as a symptom of ADHF, was prevalent in the COVID-19 cohort; while palpitations, chest pain, and cough were equally represented in both groups. Today’s primary physicians are in need more than ever of objective tools that can distinguish between different respiratory etiologies.

The main findings were:The displacement of the pleura, when quantitated by a speckled tracking tool, varies significantly between ADHF patients, COVID-19 patients, and non-respiratory patients. ADHF pleura showed larger movements than in COVID-19, while the non-respiratory patients had a relatively small displacement (Figure 3, *p* < 0.01).The velocity of the pleural movement is significantly larger for the ADHF patients than for COVID-19 pneumonia patients; while the velocity for the non-respiratory patients was very small (Figure 4, *p* < 0.01).The agreement between the physicians about the diagnosis of the LUS was poor and an objective tool is required.

Our findings by speckled tracking correlate in part with the Blue protocol of Lichtenstein et al., where ADHF is on B mode and considered to have more movement than pneumonia [21].

Our study also assessed the B-mode movement by physicians observing the pleura movement. Our results were nonconclusive, having both a very poor interrater coefficient and inconclusive results. This difference may be accounted for perhaps by varying experience between the physicians participating in our study, being a resident and an emergency medicine physician; moreover, perhaps there is an innate difficulty to assess the absence or presence of movement when there is no standard measurement to adhere to.

Our study found non-respiratory patients to have less pleural displacement by speckled tracking, when compared to COVID-19 and ADHF (Figure 2 and Figure 3). This is a novel finding in LUS literature, which to our knowledge has never compared pleural movement in varying causes of acute respiratory distress, ADHF, and pneumonia to healthy controls. This is the case when considering the proposed increase in fluid in the pleural space, be it exudative in pneumonia or transudative in ADHF, compared to the normal amount of fluid between the pleura being only 1–10 mL. Another explanation of having larger displacement in the EVLW patients may also be attributed to a function of respiratory exertion (Figure 2 and Figure 3) being highest in the ADHF, followed by the COVID-19 patients, and lastly the non-respiratory. Moreover, it seems that ADHR patients show a higher breathing exertion than COVID-19 patients (Figure 2 and Figure 3).

Ours is not the first work using speckled tracking to examine pleura. In 2019, Duclos et al. compared a B-mode LUS to speckled tracking of pleura, and found a 100% sensitivity and specificity of the longitudinal strain of pleura in diagnosing pneumothorax as compared to the B-mode ultrasound of pneumothorax; the latter had a specificity of 100%, but a sensitivity of 95% [17]. Another work by Fissore et al. found speckled tracking on LUS to improve the accuracy of the novice and intermediate sonographer in the diagnosing of pneumothorax [18].

### Limitations

The ultrasound images were collected not for the purpose of establishing pleural movement; but rather, for recording different pathological findings. There was no reference to lung zones of the patients and no homogeneity of method of saving clips. The physician performing the ultrasound recorded pathological findings. Due to the large surface area of the lung and our lack of a systemic method of recording our lung ultrasound, we are aware that pathologies may have been missed in the patients scanned. In addition, the ADHF patient population was considerably older than the COVID-19 patients as well as for the non-respiratory patients. This is an expected finding since ADHF is prevalent in an elder population, while the COVID-19 pandemic reaches all age groups. Yet we are aware this discrepancy may have influenced our results. Our results also showed large standard deviations, which is a similar finding to that in the work of Duclos et al. [17]. This may be due to the heterogeneity of the population examined, and variations of inherent properties of the pleural of the patients as well as our small sample size. Our study consisted of data from 208 clips from only sixty-five patients; values cannot be established from such and at best, trends are exhibited. Our work does raise the potential value of an objective quantification of pleural movement, which is obtained from applying speckled tracking to LUS. We hope this work may serve as a springboard to trigger other larger studies.

We believe that the trends established by our study are indicative of the need for larger studies to establish standardization. In addition, the information attained by speckled tracking may be interpreted in the context of other findings to reach the most likely cause of dyspnea.

## 5. Conclusions

In an age of the expanding use of point-of-care ultrasounds by physicians of all levels of experience, and the gaining popularity of lung ultrasounds in dealing with the COVID-19 pandemic, [10,11] there is a need to examine tools that can decrease the variability between observers in lung ultrasounds. We believe that speckled tracking of the pleura is a potential tool for objectively assessing the pleura and quantifying the extent of its movement. In addition, these findings may be extrapolated beyond COVID-19 to similar pathologies of increased EVLW, such as other pneumonias and ARDS.

With further research and the establishment of normal universal values, as well as an examination of a spectrum of values for various pathological states, speckled tracking may find its way into our toolbox in order to be used in conjunction with other established diagnostics to differentiate between various respiratory pathologies.

## Figures and Tables

**Figure 1 jcm-11-04846-f001:**
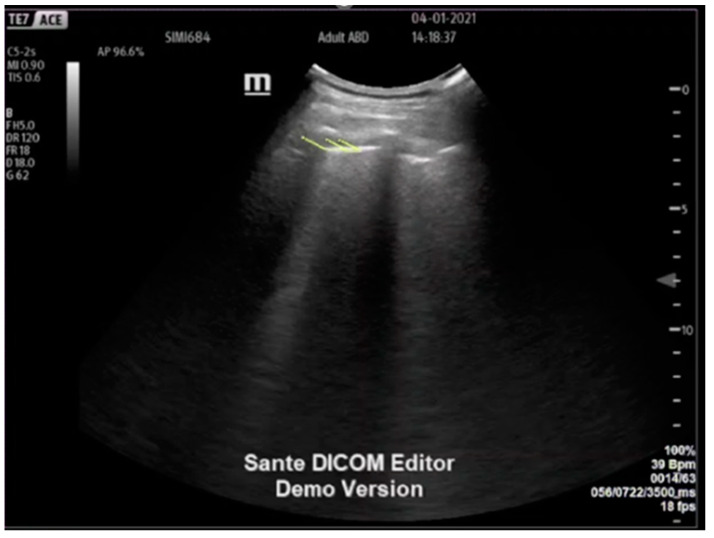
Speckled tracking applied to the B-mode ultrasound clip of the pleura of an acute decompensated congestive heart failure patient; yellow lines the vector of movement.

**Figure 2 jcm-11-04846-f002:**
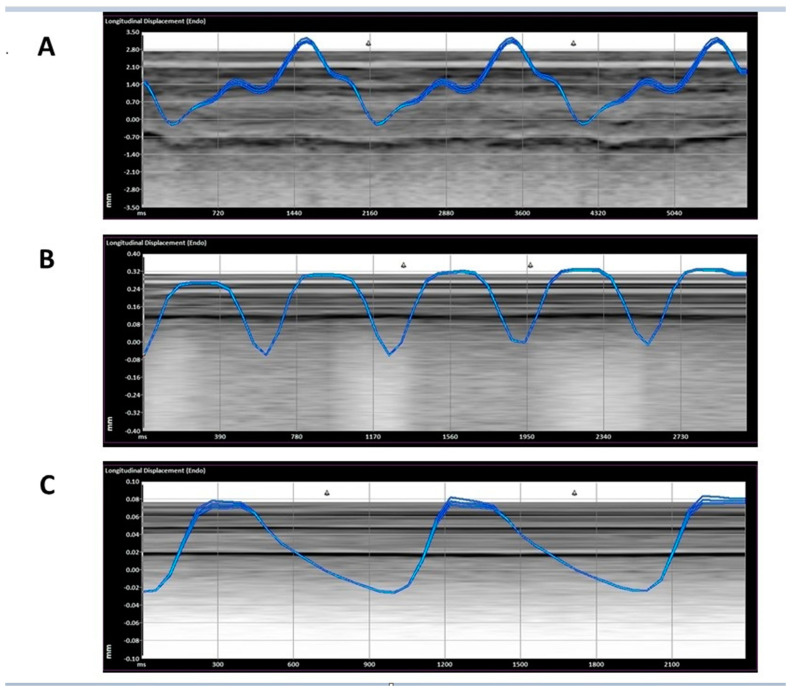
Representative measurement of the displacement. ((**A**)—ADHF (acute decompensated heart failure), (**B**)—COVID-19, (**C**)—non-respiratory).

**Figure 3 jcm-11-04846-f003:**
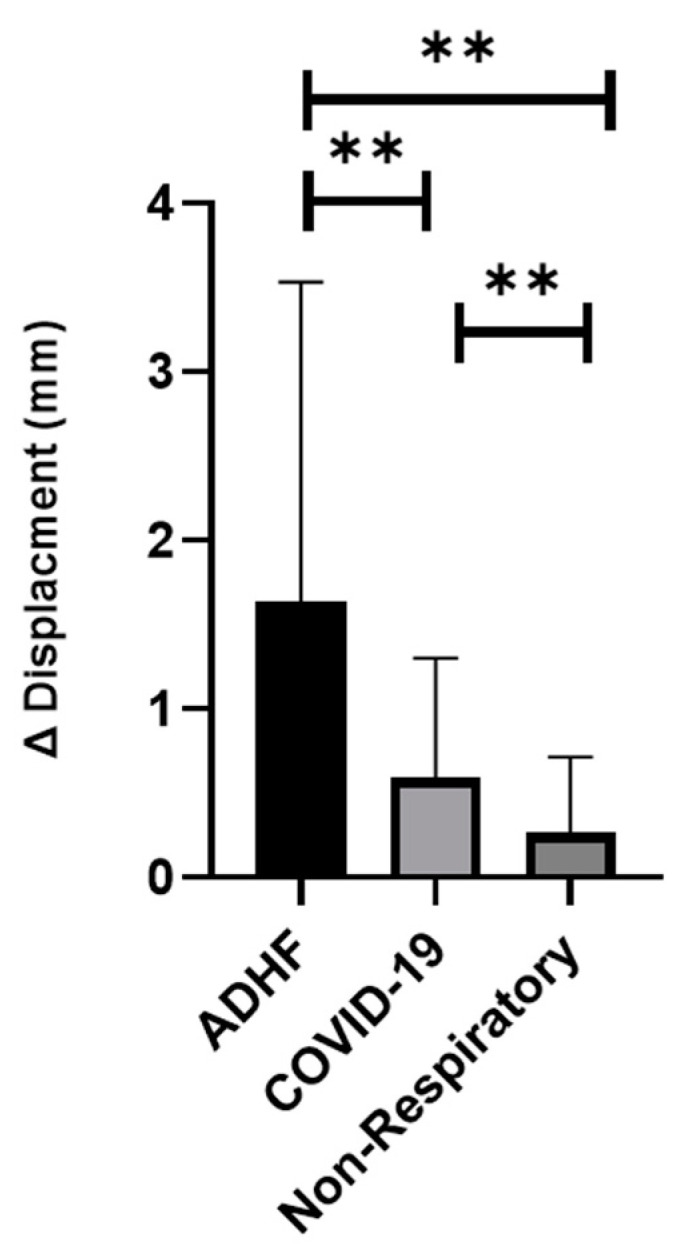
The difference between maximum and minimum displacement (mm) ADHF- (acute decompensated heart failure) (** *p* < 0.01).

**Figure 4 jcm-11-04846-f004:**
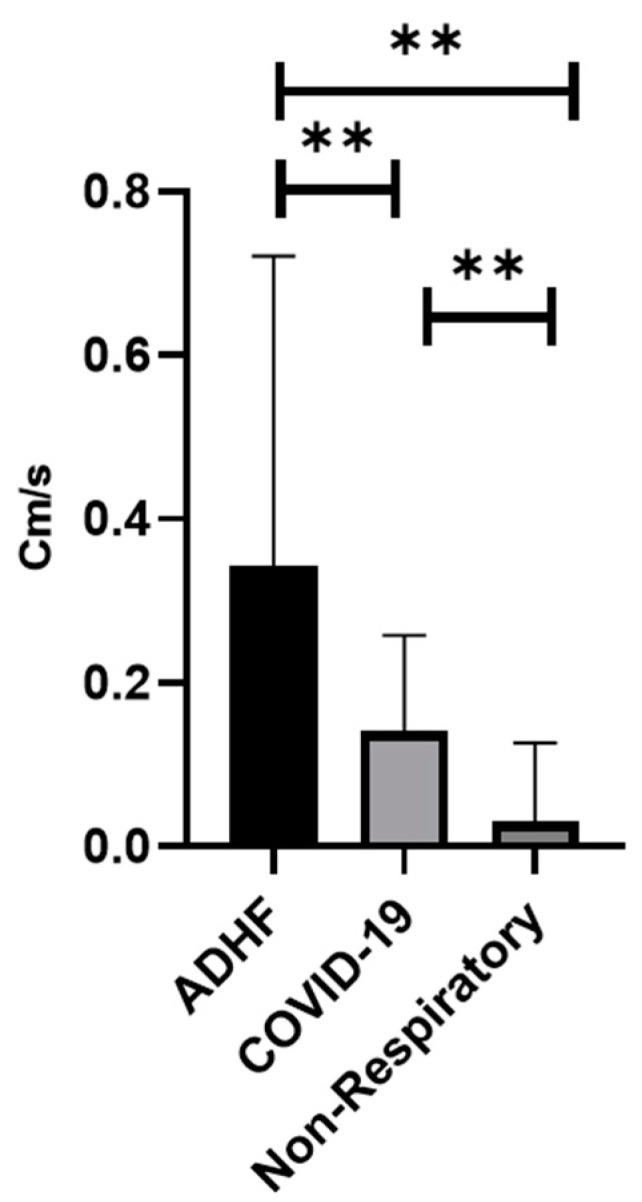
Maximum velocity (Cm/s) (** *p* < 0.01). ADHF (acute decompensated heart failure).

**Table 1 jcm-11-04846-t001:** Demographics, comorbidities and major complaints.

	ADHF	COVID-19	Non-Respiratory
N	25	21	19
LUS loops	66	60	82
M/F	6/19	10/11	13/6
age	80 ± 8	63 ± 13	51 ± 13
Comorbidities			
Congestive Heart Failure	13	2	-
Rheumatic Heart Failure	1		-
Asthma	1	2	-
Chronic obstructive pulmonary disese	4	3	-
Major complaints			
Dyspnea	19	12	4
Weakness	6	8	-
Head Injury	1	1	-
Chest Pain	6	6	3
Leg Edema	3	11	2
Fall	1	3	1
Palpitations	1	1	-
Cough	1	1	-
Hypotension	1	1	-
Electrocardiographic Changes	1	1	1
Orthopnea	1	1	-
Back Pain	1	1	1
Abdominal Pain	-	1	5
flank pain	-	-	1

ADHF—acute decompensated heart failure, LUS—lung ultrasound.

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
