# Peer review of "Speckled Tracking of Pleura—A Novel Tool for Lung Ultrasound; Distinguishing COVID-19 from Acute Heart Failure"

_jcm, 2022, doi:10.3390/jcm11164846_

Round 1

Reviewer 1 Report

I read with interest the article entitled “Speckled Tracking of Pleura- A Novel Tool for Lung Ultrasound - Distinguishing COVID-19 from Acute Heart Failure” by Tzadok et al. 

This paper explores a new research topic applied to lung ultrasound. In the last decade, and especially during the pandemic, lung ultrasound use in the emergency department and intensive care unit increased exponentially. I would suggest considering these references to implement the introduction or discussion: 

-       Meroi F, Orso D, Vetrugno L, Bove T. Lung Ultrasound Score in Critically Ill COVID-19 Patients: A Waste of Time or a Time-Saving Tool? Acad Radiol. 2021 Sep;28(9):1323-1324. doi: 10.1016/j.acra.2021.06.008. 

-       Vetrugno L, Meroi F, Orso D, D'Andrea N, Marin M, Cammarota G, Mattuzzi L, Delrio S, Furlan D, Foschiani J, Valent F, Bove T. Can Lung Ultrasound Be the Ideal Monitoring Tool to Predict the Clinical Outcome of Mechanically Ventilated COVID-19 Patients? An Observational Study. Healthcare (Basel). 2022 Mar 18;10(3):568. doi: 10.3390/healthcare10030568.

The lung ultrasound is an incredible tool in experienced hands and helps in the diagnosis and in prognosis of COVID-19 and other pathological conditions. 

With the help of this novel tool of the “Speckeld Traking of the pleura”, the authors open a new perspective on the differential diagnosis of an infective cause of increased B lines and an ADHF.

Unfortunately, the limitations of the study are several and have been correctly reported. Would you please clarify if the images have been acquired and analyzed from both ultrasound machines and both linear and curvilinear probes? Is there a bias in the results? Different ultrasound machines and different probes could give fairly discrepant results. 

Could you please specify with images and in which mode has been used for the Speckeld Traking analysis? The acquired ultrasounds are 208 but the patients are only 65, limiting the sample size to draw conclusive statements. 

I would rather stress the fact that the diagnosis could difficultly be based only on one tool, but the Speckeld Traking might be of help in specific settings and after more research and a standardized protocol. 

I would suggest revising the conclusions, without the futuristic speculation of the first paragraph, leaving it to the more concrete second paragraph. 

Author Response

Reviewer One:

This paper explores a new research topic applied to lung ultrasound. In the last decade, and especially during the pandemic, lung ultrasound use in the emergency department and intensive care unit increased exponentially. I would suggest considering these references to implement the introduction or discussion: 

-       Meroi F, Orso D, Vetrugno L, Bove T. Lung Ultrasound Score in Critically Ill COVID-19 Patients: A Waste of Time or a Time-Saving Tool? Acad Radiol. 2021 Sep;28(9):1323-1324. doi: 10.1016/j.acra.2021.06.008. 

-       Vetrugno L, Meroi F, Orso D, D'Andrea N, Marin M, Cammarota G, Mattuzzi L, Delrio S, Furlan D, Foschiani J, Valent F, Bove T. Can Lung Ultrasound Be the Ideal Monitoring Tool to Predict the Clinical Outcome of Mechanically Ventilated COVID-19 Patients? An Observational Study. Healthcare (Basel). 2022 Mar 18;10(3):568. doi: 10.3390/healthcare10030568.

I read the articles you suggested on COVID-19 and lung ultrasound, I agree they are quite relevant please see lines 54-55 with added references

Unfortunately, the limitations of the study are several and have been correctly reported. Would you please clarify if the images have been acquired and analyzed from both ultrasound machines and both linear and curvilinear probes? Is there a bias in the results? Different ultrasound machines and different probes could give fairly discrepant results. 

Thank you for this important point we needed to clarify, please see line 97-100 as to the ultrasound machine, and line 153-154 where we checked if there was a discrepancy between the use of different probes.

Could you please specify with images and in which mode has been used for the Speckeld Traking analysis?

Line 115-116, 119 figure 1

The acquired ultrasounds are 208 but the patients are only 65, limiting the sample size to draw conclusive statements. 

Agreed, an important point please see 227-231

I would rather stress the fact that the diagnosis could difficultly be based only on one tool, but the Speckeld Traking might be of help in specific settings and after more research and a standardized protocol. 

Thank you for that important point we needed to raise, please see lines 253-254

I would suggest revising the conclusions, without the futuristic speculation of the first paragraph, leaving it to the more concrete second paragraph. 

Agreed and it was deleted please see 237-242

Reviewer 2 Report

Overall well written. 
things to improve are:

- better explanation of speckle tracking lung US (also with images)

- a less “dreamy” conclusion 

- correction of the rare typos.

- comments on the advantages of speckle tracking 

   Lung US vs echocardiography. 

Author Response

Reviewer 2:

- better explanation of speckle tracking lung US (also with images)

Thank you please see line 63-66, for a more in depth explanation, and the addition of figure 1- which demonstrates the application of speckled-tracking to the B-mode clip.

- a less “dreamy” conclusion 

Thank you, understood and changed, please see lines 237-242

- correction of the rare typos.

Thank you reviewed and addressed

- comments on the advantages of speckle tracking 

Thank you addressed lines 229-231

   Lung US vs echocardiography. 

Thank you please see lines 63-64
